# The Effects of Different Processing Methods on the Levels of Biogenic Amines in Zijuan Tea

**DOI:** 10.3390/foods11091260

**Published:** 2022-04-27

**Authors:** Dandan Liu, Kang Wang, Xiaoran Xue, Qiang Wen, Shiwen Qin, Yukai Suo, Mingzhi Liang

**Affiliations:** 1Institute of Resource Plants, Yunnan University, Kunming 650500, China; liudandan@ynu.edu.cn (D.L.); 3225813775@mail.ynu.edu.cn (K.W.); xxr10221022@163.com (X.X.); 2Key Laboratory of Chemistry in Ethnic Medicinal Resources, State Ethnic Affairs Commission and Ministry of Education, Yunnan Minzu University, Kunming 650031, China; wenqiang152323@163.com; 3Tea Research Institute, Yunnan Academy of Agricultural Sciences, Menghai 666201, China; liangmingzhi@126.com

**Keywords:** biogenic amine, Zijuan tea, processing method, food quality, HPLC

## Abstract

This study aimed to evaluate the effects of processing methods on the content of biogenic amines in Zijuan tea by using derivatization and hot trichloroacetic acid extraction with HPLC-UV. The results showed that the most abundant biogenic amine in the original leaves was butylamine, followed by ethylamine, methylamine, 1,7-diaminoheptane, histamine, tyramine, and 2-phenethylamine. However, during the process of producing green tea, white tea, and black tea, the content of ethylamine increased sharply, which directly led to their total contents of biogenic amines increasing by 184.4%, 169.3%, and 178.7% compared with that of the original leaves, respectively. Unexpectedly, the contents of methylamine, ethylamine, butylamine, and tyramine in dark tea were significantly reduced compared with those of the original leaves. Accordingly, the total content of biogenic amines in dark tea was only 161.19 μg/g, a reduction of 47.2% compared with that of the original leaves, indicating that the pile-fermentation process could significantly degrade the biogenic amines present in dark tea.

## 1. Introduction

Biogenic amines (BAs), which are mainly derived from the decarboxylation of amino acids and amination/transamination of aldehydes and ketones, are small molecules of nitrogen-containing organic compounds with biological activity [1,2]. Previous studies have shown that the right amounts of BAs have positive effects on maintaining human metabolism, immune activity, and vascular activity [3,4]. However, due to the potential toxicity of BAs, excessive intake of BAs could cause toxic effects such as vomiting, headaches, and allergies, especially for sensitive people [5,6]. For instance, the intake of 8–40 mg histamine can cause slight human poisoning immediately, and severe poisoning can be caused when the value further rises above 100 mg [7]. To avoid the harmful effects of BAs, governments of various countries have formulated standards for the levels of BAs in food based on their food safety systems [6,8].

Due to the decarboxylation of amino acids by microorganisms during fermentation, the concentration of BAs in fermented foods (cheese, beer, soy sauce, etc.) is generally higher than that of non-fermented foods (vegetables, fruits, meat, etc.) [9]. For instance, several yeast species, including *Geotrichum candidum*, *Yarrowia lipolytica*, and *Debaryomyces hansenii*, have been identified as potential BA producers [10,11]. In contrast, a variety of bacteria isolated from food can effectively degrade BAs by secreting amine oxidases. Currently, strains belonging to the species *Lactobacillus plantarum*, *Lactobacillus sakei*, *Pediococcus acidilactici*, *Rhodococcus* sp., *Bacillus amyloliquefaciens*, and *Staphylococcus carnosus* have been shown to degrade BAs (e.g., histamine, tyramine, and putrescine) [12]. Hence, the microbial species have a great impact on the content of BAs in fermented foods. In addition, factors such as temperature, fermentation strain and mode, free amino acid content, salt concentration, and pH can also affect the synthesis of BAs in foods [13,14,15]. Oracz and Nebesny indicated that the content of BAs in cocoa beans increases with the increasing roasting temperature (110 °C to 150 °C) and air humidity (0.3% to 5.0%) [16]. Therefore, inhibiting the formation of BAs in foods by reducing free amino acid contents, limiting the growth of amine-positive microorganisms, and changing food processing conditions and additives has become a research focus [17,18].

To avoid serious toxic effects of BAs on human and animal bodies, it is essential to construct accurate analysis methods for the determination of BAs in various foods. As a result, various techniques, including ion chromatography with conductimetric and amperometric detection [19], gas chromatography (GC) with mass spectrometric detection [20], high-performance liquid chromatography (HPLC) [21], capillary electrophoresis (CE) [22], and electroanalytical methods [23], have been successfully used to detect BAs in foodstuffs. Among them, HPLC methods with ultraviolet (UV) and fluorescence (FL) detectors are the most frequently used methods due to their convenient operation and high sensitivity and selectivity [24].

However, when HPLC is used for detection, a derivatization step is required because most BA structures lack fluorescent or chromophore groups [24]. During the process of derivatization, the amino group of BAs can react with tagging reagents, such as 6-aminoquinolyl-N-hydroxysuccinimidy (AQC), o-phthalaldehyde (OPA), 1,2-naphthoquinone-4-sulfonate naphthalene-2,3-dicarboxaldehyde (NDA), dansyl chloride (dansyl-Cl), and N-(9-fluorenylmethoxycarbonyloxy) succinimide (Fmoc-OSu) [25,26]. Among these, dansyl-Cl is an important tagging reagent because its derivatives are relatively stable. Thus, dansyl-Cl is widely used as a pre-column derivatization reagent for the determination of BAs [27].

Zijuan tea plant, which has purple leaves, buds, and stems, is a cultivar bred from an individual plant of Yunnan Daye (*Camellia sinensis* var. *assamica*) by the Yunnan Academy of Agricultural Sciences, China [28]. Due to its richness in anthocyanins, multiple health benefits, and unique taste, Zijuan tea is welcomed by a vast number of consumers despite its high price [29,30]. The leaves of the Zijuan tea plant can be processed into Zijuan green tea by fixation, rolling, sun (machine) drying, and without high humidity storage or microbial fermentation [29]. Furthermore, Zijuan green tea can be made by microbial fermentation into the more valuable fermented teas, which can be classified as white tea, black tea, and dark tea according to the degree of fermentation [31]. During the process of fermentation, the amino acids of Zijuan tea are degraded by microorganisms, indicating that there may be a certain amount of BAs in fermented Zijuan tea. Unfortunately, the levels of BAs in Zijuan tea have not been investigated.

The purpose of this study was to evaluate the effects of different processing methods on the levels of BAs in Zijuan tea. First, after extraction from Zijuan tea (original leaves, green tea, white tea, black tea, and dark tea) and derivatization by dansyl chloride, the composition of BAs was determined by using high-performance liquid chromatography (HPLC). Then, the types and contents of BAs in Zijuan tea were analysed to evaluate the effects of different processing methods on the formation of BAs.

## 2. Materials and Methods

### 2.1. Samples

The fresh leaves of Zijuan tea were provided by a tea plantation in Pu’er city, Yunnan Province, China. After spreading and drying for 6–8 h (~80% humidity), the tea leaves were made into green tea by de-enzyming (~200 °C for 2–3 min), rolling, and air drying (~6% humidity). To prepare white tea, the fresh leaves were withered at 21–25 °C for 48 h (~20% humidity), and then the withered leaves were dried to 6% humidity at 80 °C. The manufacturing process of black tea was as follows: the tea leaves were withered at 28 °C for 12–16 h to reach ~60% humidity. After rolling and fermentation (28 °C for 5–6 h), the leaves were dried at 80 °C for 1–2 h to obtain black tea with ~6% moisture. To obtain dark tea, the leaves were treated by de-enzyming (~200 °C for 2–3 min), rolling, and pile-fermentation (22–25 °C for 30–40 days). Then, the fermented leaves were dried at 80 °C until the humidity reached approximately 6%.

### 2.2. Chemicals

Histamine (HIM, dihydrochloride, assay: >99%), pyrrolidine (PYN, assay: 99%), tyramine (TYM, hydrochloride, assay: >98%), methylamine (MTM, hydrochloride, assay: >98%), ethylamine (ETM, hydrochloride, assay: 98%), putrescine (PTN, dihydrochloride, assay: >98%), butylamine (BTM, assay: 99.5%), 1,7-diaminoheptane (DIM, assay: 98%), 2-phenethylamine (PEM, hydrochloride, TraceCERT^®^), dansyl chloride (assay: ≥99.0%), and trichloroacetic acid (assay: ≥99.0%) were purchased from Sigma (Merck, Darmstadt, Germany). Chromatographic methanol, formic acid, and acetonitrile were obtained from Merck.

### 2.3. Sample Preparation

After drying for 24 h at 45 °C, the original leaves and Zijuan tea were minced and screened with a 100 mesh sieve. The powder of original leaves and Zijuan tea was combined with hot trichloroacetic acid (5% *w*/*v*) with a solid–liquid ratio of 1:10 (*w*/*v*). The resulting mixture was treated with ultrasound (40 Hz) for 20 min and centrifuged at 8000 rpm for 10 min to obtain the supernatant. Then, the residuals were extracted twice by using the above method. Finally, the three extraction supernatants were mixed to obtain the BA extracts.

The derivatization of BAs followed the method described by Restuccia et al. with slight modifications [32]. Dansylation reactions were performed in a 10 mL tube by adding 0.5 mL of BA extract (or standard solution), 2.0 mL of Dns-Cl solution (10.0 mg/mL in acetone), and 1.0 mL of sodium carbonate-sodium bicarbonate buffer (pH 10.0). After reacting at 60 °C for 15 min, 100 μL of ammonium hydroxide (25% *v*/*v*) was added to remove the excess Dns-Cl. The reaction solution was filtered with 0.22 μm membrane filters and then used for HPLC analysis.

### 2.4. HPLC Chromatographic Conditions

The determination of BAs was carried out by using a 1260 Infinity II HPLC (Agilent, Santa Clara, CA, USA) consisting of a UV detector, a quaternary pump, and HP ChemStation software. For chromatographic separation of BAs, a ZORBAX Eclipse Plus C_18_ column (4.6 mm × 250 mm, 5 μm) with a C_18_ guard-pak (4.6 mm × 10 mm, 5 μm, Agilent, Santa Clara, CA, USA) was used. The mobile phase consisted of purified water (A) and methanol/acetonitrile (50:50, B). The gradient program was 30% A from 0–7 min, 30–25% A from 7–10 min, 25–0% A from 10–23 min, hold at 0% A from 23–29 min, 0–30% A from 29–30 min, and post run at 30% A for 10 min. The flow rate and injection volume were set as 0.5 mL/min and 10 μL, respectively.

### 2.5. Preparation of Amine Standard Solutions

To prepare standard solutions of BAs, the nine BAs were severally dissolved in 5% trichloroacetic acid at a concentration of 2.0 mg/mL. Then, different aliquots of each amine solution were taken to prepare a standard mix of Bas and diluted with 5% trichloroacetic acid to a volume of 5 mL. The concentrations of each amine in the mixture were 0.1, 0.2, 0.5, 1, 2, 5, 10, 25, and 50 μg/mL. BAs of Zijuan tea were recognized by comparing the retention time of peaks with those of amine standard solutions. 

### 2.6. Validation of the BA Determination Method

Solutions containing nine BAs were prepared at concentrations of 3, 15, and 30 mg/mL. Then, 100 μL of solution was added to 10 g of Zijuan tea. The BA content of samples added BAs and control samples (Zijuan tea without BAs) were detected by using HPLC.

### 2.7. Statistical Analysis

Unless otherwise specified, all the detections of BAs in Zijuan tea were repeated three times, and the mean values were displayed with relative standard deviations (RSD). Analysis of variance (one-way ANOVA) was carried out by using SPSS Version 12.0 software, and the significance was evaluated with the value of *p* < 0.05. 

## 3. Results and Discussion

### 3.1. Method Performances

During the process of de-enzyming and microbial fermentation, the amino acids in tea leaves may be decarboxylated by high temperature or microbial decarboxylase to generate BAs. To evaluate the risk of BAs, several studies have tried to use HPLC-UV to determine the contents of BAs in teas [33,34]. For instance, Shen et al. determined the content of methylamine, ethylamine, putrescine, spermidine, histamine, tryptamine, cadaverine, and tyramine in Pu-erh teas, and the results indicated that the main BAs in Pu-erh teas are methylamine, ethylamine, and tryptamine [33]. However, the effects of different processing methods on the formation of BAs in tea have not been studied. In the present work, after extraction and derivatization (Dns-Cl), the BAs in Zijuan tea (original leaves, green tea, and fermented teas) were detected using HPLC. Based on knowledge about BAs in tea and other foods, nine BAs, MTM, ETM, BTM, PEM, PTN, PYN, DIM, HIM, and TYM, were identified in Zijuan tea by comparison with their standard substances. 

As shown in Figure 1, nine types of BAs were perfectly separated within 30 min, while quantification of BA content with a low level could be achieved. Calibration curves of BAs were obtained by analysing the peak area of each analyte at nine different concentrations, and each concentration of analyte was detected three times. The results of HPLC detection showed that the correlation coefficients of the calibration curve for the analysed BAs were higher than 0.995, indicating that the measurements of the nine BAs displayed good linear relationships within the range of 0.1–50 μg/mL (Table 1). Limits of detection (LODs) and limits of quantitation (LOQs) were evaluated by considering signal-to-noise ratios of 3 and 10, respectively. The LODs of BAs, which were evaluated by detecting the standards at low concentration levels, were 0.15–0.96 μg/mL, while the LOQs were 0.50–3.22 μg/mL. The recovery percentages, evaluated by comparing the amount added and the amount obtained after spiking, ranged from 92.2% to 104.4%, and the relative standard deviation (RSD) was in the range of 0.9% to 4.7% (Table 2). These results indicated that the method of extraction, derivatization, and determination is repeatable and feasible. 

### 3.2. Levels of BAs in Original Leaves and Green Tea

As low-molecular-weight bioactive substances, BAs such as spermine and putrescine have been proven to be widely present in fresh fruits and vegetables [35]. To evaluate the effects of different processing methods on the levels of BAs, the content of BAs in the original leaves of Zijuan tea was first determined. As shown in Figure 2, seven types of BAs (MTM, ETM, BTM, PEM, DIM, HIM, and TYM) were detected in the original leaves. Among them, the concentrations of MTM, ETM, and BTM were relatively high, reaching 58.68 μg/g, 68.23 μg/g, and 78.87 μg/g, respectively. The concentrations of HIM and TYM, which have been shown to often be present together in plants, were 27.72 μg/g and 26.22 μg/g, respectively [36]. Furthermore, the PEM (8.31 μg/g) and DIM (37.23 μg/g) were also detected in the original leaves. 

After spreading, de-enzyming, rolling, and air drying, the original leaves of Zijuan tea can be made into green tea, which was found to be rich in phenolic compounds and antioxidants [37]. During this manufacturing process, the high-temperature treatment (~200 °C for 2–3 min) for de-enzyming may have a significant effect on the formation of BAs in green tea. The results of BA detection showed that the contents of ETM, PEM, and HIM in green tea reached 516.07 μg/g, 25.85 μg/g, and 73.22 μg/g, increased by 656.4%, 211.1%, and 164.2% compared with those of the original leaves, respectively (Figure 2 and Appendix A). Similarly, previous studies have also shown that the levels of BAs such as PEM, HIM, and TYM in coffee and cocoa beans would significantly increase after high-temperature treatment [16,38]. Meanwhile, two new BAs, PYN (47.03 μg/g) and PTN (36.14 μg/g), were synthesized in the process of producing green tea. In contrast, the content of MTM in green tea was reduced by 67.55% compared with that of the original leaves (Figure 2). Finally, the total content of BAs in green tea reached 868.14 μg/g, an increase of 184.4% compared with that of the original leaves (305.26 μg/g). 

### 3.3. Levels of BAs in Fermented Teas

At present, multiple studies have shown that microbial fermentation has two effects on the content of BAs in foods. The decarboxylase of microorganisms can catalyze the decarboxylation of amino acids in food to generate corresponding BAs [9,39]. However, a number of microorganisms can produce amine-degrading enzymes, such as monoamine oxidase and diamine oxidase, to degrade BAs in foods [3,40]. Hence, the content of BAs in fermented foods is largely determined by the types of microorganisms used for fermentation. Due to concerns about the accumulation of BAs during the fermentation process of Zijuan tea, the levels of BAs in white tea (slightly fermented tea), black tea (fermented tea), and dark tea (post-fermented tea) were also determined by HPLC. 

As shown in Figure 3, eight BAs were detected in white tea. Among them, the contents of ETM, PEM, DIM, and HIM reached 550.18 μg/g, 31.86 μg/g, 44.64 μg/g, and 50.30 μg/g, increased by 706.4%, 283.4%, 19.9%, and 81.5% as compared with those of the original leaves, respectively. Meanwhile, the contents of MTM and TYM were reduced by 64.1% and 81.3%, respectively. Previous studies have shown that hot air drying at a relatively low temperature (~80 °C) did not significantly affect the formation of BAs [41]. To confirm this problem, the effect of air drying on the BA content of Zijuan tea leaves was also investigated in this study, and the results indicated that the treatment at 80 °C for 1–2 h could not significantly affect the levels of BAs (data not shown). However, white tea is made by the treatment of withering and air drying, indicating that the physiological activities of microorganisms used in withering have a certain impact on the formation of BAs. After withering, the original leaves of Zijuan tea can be further fermented to obtain black tea. The contents of MTM, PEM, HIM, and TYM in black tea were 10.21 μg/g, 14.03 μg/g, 24.38 μg/g, and 8.78 μg/g, which were reduced by 51.5%, 56.0%, 51.5%, and 39.3% compared with those of white tea, respectively (Figure 3 and Appendix A). PYM and BTM were not detected in the black tea. Conversely, the concentration of ETM increased to 757.59 μg/g, indicating that the microorganisms employed for black tea fermentation can catalyze the synthesis of ETM. 

As a post-fermented tea with special sensory characteristics, dark tea can be produced by the treatment of de-enzyming, rolling, pile-fermentation, and air drying [42]. Hence, the levels of Bas in dark tea are affected by both high temperature and microbial fermentation. Surprisingly, the concentrations of MTM, ETM, BTM, and HIM were 23.60 μg/g, 19.63 μg/g, 44.37 μg/g, and 16.90 μg/g, reduced by 59.8%, 71.2%, 43.7%, and 39.0% compared with those of the original leaves, respectively (Figure 3 and Appendix A). Meanwhile, TYM was not detected in dark tea. Only the content of PEM was increased by 216.6% compared with that of the original leaves. Due to the degradation of ETM, MTM, BTM, HIM, and TYM, the total content of Bas in dark tea was only 161.19 μg/g, which was reduced by 47.2% compared with that of the original leaves.

For the manufacturing process of Zijuan white tea and black tea, the treatment of withering significantly promoted the formation of Bas, especially ETM (Figure 3). Previous studies showed that the proteins of Zijuan tea leaves were hydrolyzed into amino acids (mainly leucine, isoleucine, and tyrosine) during withering, which provide the raw material for the synthesis of Bas [43]. After the fermentation process of black tea, the content of ETM was further increased to 757.97 μg/g. Hence, inhibiting ETM synthesis during withering and fermentation is essential to reduce the BA content of white tea and black tea. By comparing the manufacturing process of green tea and dark tea, it was found that the two processes are similar, except that dark tea needs to be pile-fermented. However, the BA content of dark tea is only 18.6% of that of green tea, indicating that the microorganisms used in pile-fermentation have a good ability to degrade Bas. Due to the long period of the pile-fermentation (≥30 days), the microbial composition of different fermentation stages will show obvious differences. For instance, Xu et al. indicated that microorganisms in early pile-fermentation stage were mainly bacteria, and Saccharomycetes and Mycetes played an important role in the middle and late stages [44]. To isolate biogenic amine-degrading strains, we will reveal the relationship between BA content and microbial species by microbial diversity analysis in future work. 

## 4. Conclusions

Zijuan tea, a product rich in anthocyanins, can be made into green tea, white tea, black tea, and dark tea with various health benefits and unique taste. However, the treatment of high temperature and microbial fermentation for producing green tea and fermented teas may significantly affect the synthesis of Bas. To clarify the effects of different processing methods on the formation of Bas, the BA contents of green tea and fermented teas were determined by HPLC-UV. Due to the accumulation of ETM, PEM, and HIM, the BA content of green tea increased by 184.39% compared with that of the original leaves, indicating that the manufacturing process of green tea, especially the high temperature treatment, can significantly enhance the synthesis of BAs.

For white tea and black tea, the withering and fermentation treatments significantly enhanced the formation of ETM, which directly led to the BA content of white tea and black tea reaching 822.14 μg/g and 850.82 μg/g, respectively. Unexpectedly, the total content of BAs in dark tea was only 161.19 μg/g after treatment with high temperature, fermentation, and air drying, indicating that the microorganisms used in the pile-fermentation of dark tea have excellent degrading ability for BAs. To avoid the accumulation of BAs in teas, it is important to further reveal the degradation mechanism of BAs during dark tea fermentation. 

## Figures and Tables

**Figure 1 foods-11-01260-f001:**
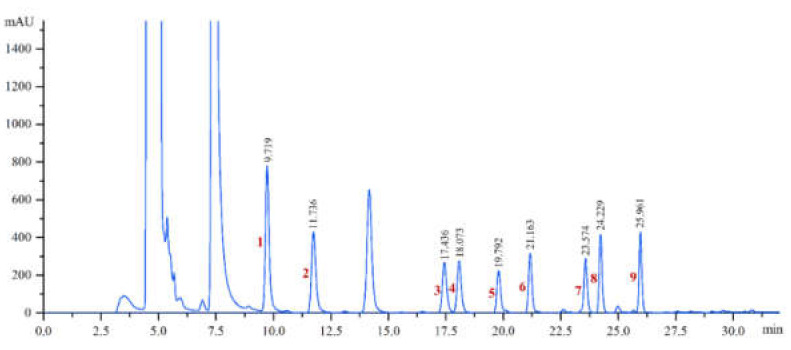
HPLC chromatogram map of the mixed biogenic amine standard (50 μg/mL each). Peak assignment: 1, MTM; 2, ETM; 3, PYN; 4, BTM; 5, PEM; 6, PTN; 7, DIM; 8, HIM; and 9, TYM.

**Figure 2 foods-11-01260-f002:**
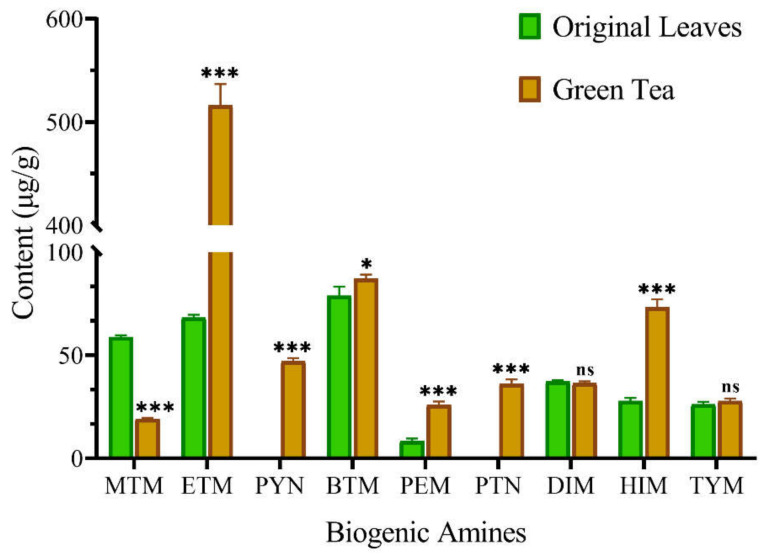
Changes in the contents of biogenic amines in the original leaves and green tea. Each column indicates the mean ± SD of the content of a biogenic amine. The significant differences in the contents of biogenic amines between the original leaves and green tea were analyzed using Student’s *t*-test for independent samples. * *p <* 0.05, *** *p <* 0.001, ns no significance.

**Figure 3 foods-11-01260-f003:**
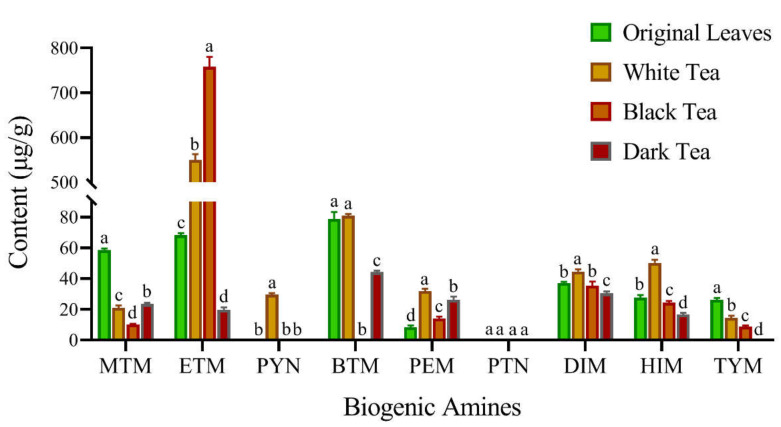
Changes in the contents of biogenic amines in the original leaves and fermented teas. Each column indicates the mean ± SD of the content of a biogenic amine. The significant differences in the contents of biogenic amines among the original leaves, white tea, black tea, and dark tea were analysed using the Duncan multiple mean comparison test. Columns labelled with the same letter had no significant difference among the four samples, *p* > 0.05.

**Table 1 foods-11-01260-t001:** The performances of LC-UV method.

BAs	Calibration Curve Equation	R^2^	LOD (μg/mL)	LOQ (μg/mL)
Methylamine	y = 228.7x + 82.568	0.9995	0.15	0.50
Ethylamine	y = 113.73x + 115.19	0.9999	0.31	1.04
Pyrrolidine	y = 142.42x − 18.254	0.9999	0.29	0.96
Butylamine	y = 71.14x + 148.51	0.9951	0.42	1.40
2-Phenethylamine	y = 64.163x + 5.8879	0.9998	0.44	1.51
Putrescine	y = 53.625x + 91.609	0.9996	0.33	0.98
1,7-Diaminoheptane	y = 35.049x − 18.118	0.9983	0.96	3.22
Histamine	y = 73.276x + 7.4512	0.9999	0.43	1.46
Tyramine	y = 64.951x + 47.487	0.9995	0.36	1.20

**Table 2 foods-11-01260-t002:** Recovery rate and relative standard deviation (RSD) of the nine BAs.

BAs	Recovery Rate (%)			RSD (%)		
	30 μg/g	150 μg/g	300 μg/g	30 μg/g	150 μg/g	300 μg/g
Methylamine	98.2	96.5	100.2	3.5	2.3	1.9
Ethylamine	97.5	105.1	95.7	2.8	1.5	1.3
Pyrrolidine	96.8	92.2	103.4	1.6	1.1	2.4
Butylamine	99.2	94.5	95.7	4.2	3.4	4.7
2-Phenethylamine	97.1	99.7	95.9	1.3	0.9	2.8
Putrescine	93.8	96.9	98.4	1.8	2.4	4.1
1,7-Diaminoheptane	99.2	95.8	97.5	2.2	1.2	3.8
Histamine	95.9	97.4	94.6	1.0	2.3	1.7
Tyramine	99.1	104.4	98.7	2.5	1.4	2.7

## Data Availability

Available data are presented in the manuscript.

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
