# Peer review of "The Effects of Different Processing Methods on the Levels of Biogenic Amines in Zijuan Tea"

_foods, 2022, doi:10.3390/foods11091260_

Round 1
Reviewer 1 Report
Foods’s manuscript
General comments:
In general, the manuscript titled
The manuscript titled “effects of different processing methods on levels of bio genic amines in Zijuan teas” has a valuable topic and scientifically sounds. The manuscript is well written. The English language and style are fine except for moderate English language check required. The experimental design is adequate.
There are some MINOR comments.
Detailed comments:
In general, please avoid using personal pronouns such as line 230 (our study) and apply this rule throughout the manuscript.
Title:
Please change Zijuan teas to Zijuan tea and apply this in through the manuscript.
Keywords:
The keywords list was carefully and accurately chosen.
Abstract:
The aim of the study and the main objectives were not clearly stated.
Please state the aim of this study clearly in this section.
Introduction:
This section didn’t provide enough background about the topic. The introduction needs to be elongated and enriched.
Materials and Methods:
The experimental design is adequate and suitable to the current study.
Results and Discussion:
This section is well written, and the data is well presented.
Conclusion:
This section is ok. This section provides a good conclusion for the study and includes the significant findings with some recommendations for further study about this point.
References:
The authors provided enough citations, and it was UpToDate.
* I am convinced that this manuscript will be suitable for publication in foods journal after MINOR revision.
Reviewer 2 Report
This article aims to evaluate the effect of processing in Zijuan tea on certain biogenic amines. It is true that are no articles that determine these amines in this type of food. However, the reason for this fact is not discussed. Maybe, because the risk associated with these components could be low, in any case, after the results are obtained, it would be interesting to evaluate the risk associated with these amines when consuming Zijuan tea as an infusion.
Regarding the manuscript, although it is well written and organized. However, maybe there is unnecessary information, such as in Figure 1 (the name of Tym peak has been not included) and Table 2. Similar HPLC methods for determining biogenic amines have been validated in other matrices. In addition, there is repeated information in the text, Figures and Tables, for example, Table 3 and Figures 2 and 3.
Some paragraphs correspond more to an introduction than a discussion in the results and discussion section. For example, see lines 133-141 or 206-214
Reviewer 3 Report
Foods
Article
The effects of different processing methods on levels of biogenic amines in Zijuan teas
Seems to be an effective work from China on the biogenic amines (BAs), mainly derived from the decarboxylation of amino acids and amination/transamination of aldehydes and ketones, with biological activity.
The authors have evaluated the effects of different processing methods on the levels of BAs in Zijuan teas. Firstly, after extraction from Zijuan teas (original leaves, green tea, white tea, black tea, and dark tea) and derivatization by dansyl chloride, the composition of BAs was determined by using high-performance liquid chromatography (HPLC). Then, the types and contents of BAs in Zijuan teas were analyzed to evaluate the effects of different processing methods on the formation of BAs.
Simple but tedious work,
Well organized, but with several grammatical errors
These must be corrected before going for publications
All figures and tables were adequate, figures were extremely good,
The conclusions were ok, maybe a native English speaker should read this before submitting the corrections.
